# Objective Structured Assessment of Debriefing (OSAD) in simulation-based medical education: Translation and validation of the German version

**Sandra Abegglen**[1]*, **Andrea Krieg**[2], **Helen Eigenmann**[1], **Robert Greif**[2,3]

**1** Department of Health Psychology and Behavioural Medicine, Institute of Psychology, University of Bern, Bern, Switzerland, **2** Department of Anaesthesiology and Pain Therapy, Bern University Hospital, University of Bern, Bern, Switzerland, **3** School of Medicine, Sigmund Freud University Vienna, Vienna, Austria

* sandra.abegglen@psy.unibe.ch

**Data Availability Statement:** All datasets are available from the figshare database (https://doi.org/10.6084/m9.figshare.13061555.v1).

## Abstract

Debriefing is essential for effective learning during simulation-based medical education. To assess the quality of debriefings, reliable and validated tools are necessary. One widely used validated tool is the Objective Structured Assessment of Debriefing (OSAD), which was originally developed in English. The aim of this study was to translate the OSAD into German, and to evaluate the reliability and validity of this German version (G-OSAD) according the 'Standards of Educational and Psychological Measurement'. In Phase 1, the validity evidence based on content was established by a multistage cross-cultural adaptation translation of the original English OSAD. Additionally, we collected expert input on the adequacy of the content of the G-OSAD to measure debriefing quality. In Phase 2, three trained raters assessed 57 video recorded debriefings to gather validity evidence based on internal structure. Interrater reliability, test-retest reliability, internal consistency, and composite reliability were examined. Finally, we assessed the internal structure by applying confirmatory factorial analysis. The expert input supported the adequacy of the content of the G-OSAD to measure debriefing quality. Interrater reliability (intraclass correlation coefficient) was excellent for the average ratings (three raters: ICC = 0.848; two raters: ICC = 0.790), and good for the single rater (ICC = 0.650). Test-retest reliability was excellent (ICC = 0.976), internal consistency was acceptable (Cronbach's α = 0.865), and composite reliability was excellent (ω = 0.93). Factor analyses supported the unidimensionality of the G-OSAD, which indicates that these G-OSAD ratings measure debriefing quality as intended. The G-OSAD shows good psychometric qualities to assess debriefing quality, which are comparable to the original OSAD. Thus, this G-OSAD is a tool that has the potential to optimise the quality of debriefings in German-speaking countries.

## Introduction

Simulation-based medical education (SBME) has gained increasing importance for learning of patient care and safety in healthcare over the last two decades [1, 2]. Research supports its general effectiveness for the improvement of learner knowledge [2], skills enhancement [2, 3], and

**Funding:** The Department of Anaesthesiology and Pain Medicine, Bern University Hospital, Bern, Switzerland (http://www.anaesthesiologie.insel.ch/de/research/) granted a departmental research grant (GRRD-1-19) to RG. No other external funding was obtained. The funders had no role in study design, data collection and analysis, decision to publish, or preparation of the manuscript.

**Competing interests:** The authors have declared that no competing interests exist.

behavioural changes in the clinical settings [2, 4], as well as for patient-related outcomes [2, 4]. The subsequent debriefing is an integral component of learning by simulation [5–7].

This debriefing is defined as a guided discussion in which two or more people purposefully interact with each other to reflect and analyse their emotional states, thoughts and actions during and after the simulations. An effective debriefing provides insights from the experience, and allows application of the lessons learned to future medical practice, to improve performance [6, 8, 9]. Experiential learning recognizes that the active role of the participants during this reflection of their experience to generate new knowledge is important for the learning process itself. The instructor guides this process and supports the participants in making sense of the events they have experienced [5, 10, 11]. The facilitation of high-quality debriefing to maximise this learning has been identified as a challenge, and indeed an art, that requires continuous education [5, 6, 10, 11].

Debriefing improves medical knowledge and technical and non-technical skills [12, 13], and can enhance performances by ~25% [14]. In contrast, SBME without debriefing offers lesser benefit to the participants [15, 16]. Despite the essential role of debriefing for learning in SBME, more research on the quality and efficacy of debriefing has been claimed to optimise the efficacy of the debriefing [14, 17]. Tools to assess the quality of debriefing have been developed [17, 18]. One of these tools is the Objective Structured Assessment of Debriefing (OSAD), which was developed in English in the UK, based on evidence and end-user opinion [17]. The OSAD comprises eight categories of effective high-quality debriefing, which are anchored to a behavioural rating scale. The OSAD serves several purposes: (1) to assess the quality of debriefing; (2) to provide feedback to instructors; (3) to enable relevant research; (4) to guide novice instructors and clinical trainers; (5) to exchange best practices; and (6) to promote high standards in debriefing [17, 19].

No German version of the OSAD has been devised for simulation research and debriefing in SBME. Therefore, this study translated the original English version of the OSAD into German and analysed its psychometric properties in German, to create the first validated German version of the OSAD (G-OSAD).

## Methods

The eight categories of high-quality debriefing of the original English OSAD [17] include approach of the facilitator, establishment of a learning environment, learner engagement, gauging learner reaction, descriptive reflection, analysis of performance, diagnosis of performance gaps, and application to future clinical practice. A 5-point scale (1, minimum; 5, maximum) was used to rate the performance of the instructor for each category, which resulted in a global score from 8 (minimum) to 40 (maximum). The supporting evidence for its use has been provided through simulations and testing in clinical practice, and includes the face, content and concurrent validity, the interrater reliability, the test-retest validity, and the internal consistency [17, 19].

We used the framework of validity described in the 'Standards of Educational and Psychological Measurement' [20] and collected validity evidence elements based on the 'content' and the 'internal structure' in two phases. In Phase 1, validity evidence was gathered based on the content by translation, back-translation and cross-cultural adaptation, to ensure consistency between the original English OSAD and the G-OSAD developed here. Additionally, we obtained expert input on whether the content of the G-OSAD was an adequate representation of the construct quality of debriefing. In Phase 2, video recorded debriefings of SBME were rated using the original English OSAD and the G-OSAD, to collect validity evidence based on

the internal structure and to compare the ratings of the original English OSAD and the G-OSAD developed.

## Phase 1: Validity evidence based on content

A multi-stage process based on the translation, review, adjudication, pre-testing and documentation (TRAPD) methodology [21] guided the translation and cross-cultural adaptation process, with each stage documented. This was based on the following processes:

1. Three psychologists, three medical practitioners and three experienced simulation instructors each individually translated the original English OSAD into German.

2. Each of these professional groups then agreed upon a consensus version.

3. One person in each of the professional groups (authors AK, RG, HE) finally agreed on a German version in a consensus meeting.

4. Two bilingual speakers with medical backgrounds (German/English native: MB, FU [Acknowledgements]) independently back-translated this German version to English for a consensus back-translation. Both were blinded to the original English OSAD.

5. An expert committee consisting of two psychologists (authors SA, HE), one physician (author AK) and one native English speaker (MB, Acknowledgements) compared the consensus back-translation to the original OSAD. Where there were differences, the German version was adapted to match the meaning of the original English OSAD.

6. This German version, together with a semi-structured questionnaire, was sent to nine expert instructors of SBME (3 physicians (Bern University Hospital and Cantonal Hospital Luzern), 3 nurses (Bern University Hospital), 2 paramedics (Bern University Hospital, Swiss Institute of Emergency Medicine Nottwil), 1 midwife (University of Applied Health Sciences Bern); 67% men; mean age: 48.56 ±8.32 years; experience with SBME: 124.11 ±153.94 courses; working experience: 21.56 ±8.86 years). Five of them were familiar with the original English OSAD. The semi-structured questionnaire included open questions regarding the view of the participants, the comprehensibility of the instructions for use, further suggestions for improvement, and the question: "How well can an instructor's ability to conduct a debriefing be assessed by the German OSAD version?" (Likert-scale: 1, extremely poor; 10, extremely well).

7. The expert committee (see point 5) then integrated the results from the semi-structured questionnaire into the final version of the G-OSAD (S1 Appendix).

This translation and the expert input constituted the validity evidence elements based on content.

## Phase 2: Validity evidence based on internal structure

With written informed consent, 57 debriefings (duration, 20–59 min.) were video recorded to apply the G-OSAD. These debriefings were from SBME courses for anaesthesia residents and nurses at the Bern University Hospital Simulation Centre (Bern, Switzerland). Each debriefing was co-led by two instructors, who were previously trained according to EuSim-Group educational standards [22] (experienced simulation instructor trainers who provide simulation instructor courses in collaboration with medical and research partners). The instructor demographics were recorded.

Three psychology students received rater training, that comprised: (a) study of and discussions around the G-OSAD; (b) discussions of potential observer bias; (c) definition of technical and non-technical medical terms; (d) pilot rating of six video recorded debriefings, followed by discussions to establish common understanding of the rating categories. The rater training was conducted and supervised by an experienced psychologist (SA) with four years of experience with debriefings in SBME. These newly trained raters evaluated the overall quality of debriefings conducted by the two simulation instructors by first watching the video recorded debriefings and taking notes, and then filling in the G-OSAD. After the rater training, each of them rated all of the 57 video recorded debriefings in a random order. Two months later, the same three raters scored six debriefings again, randomly selected from the original 57 (i.e., ~10% of the original sample), to establish test-retest reliability. Additionally, three different psychology students rated the same 57 debriefings using the original English OSAD, to compare these with the G-OSAD ratings (datasets available from [23]).

The raters were not able to score the second category of 'establishes learning environment'. Each SBME course compromises of an introduction (establishing a learning climate, clarifying the expectations and objectives from the learner, opportunity to familiarize with the simulation environment), followed by three simulation scenarios with debriefing. At the Bern University Hospital Simulation Centre the simulation scenarios are videotaped to facilitate debriefings, whereas the introduction of the SBME course is not video recorded. Therefore, this category could not be assessed.

## Statistical analysis

Statistical significance was set at $p < 0.05$ for all analyses. The mean scores and standard deviations were calculated for the Likert-scale questions of the semi-structured questionnaire on the content of the G-OSAD.

The intraclass correlation coefficients (ICCs; two-way random model, absolute agreement type) were calculated as the measures for interrater reliability for the total G-OSAD and for each category (except 'establishes learning environment'; see above). The ICCs of both the single measures (ratings of one rater) and the average measures (means of the raters) were computed for comparisons of their reliability, and thereby to derive possible implications for the use of the G-OSAD.

The ICC (consistency type) of the average total G-OSAD ratings with the average total original English OSAD ratings was calculated as the measure of agreement between the ratings of the two versions. To estimate test-retest reliability, the ICC (absolute agreement type) was calculated based on the average total scores of the randomly selected 10% G-OSAD ratings.

Exploratory factor analysis was performed, followed by confirmatory factor analysis (CFA), based on the average ratings of the G-OSAD categories (excluding 'establishes learning environment'; see above). The adequacy of the observed data was evaluated using the Kaiser–Meyer–Olkin criterion and Bartlett's test of sphericity. The data analysis was conducted using the R-packages *lavaan* [24], *psych* [25] and *simsem* [26], in the R statistical language [27]. Preliminary analyses indicated violation of the assumption of multivariate normality of the G-OSAD items (Mardia's coefficient of skewness = 153.58; $p < 0.001$). Thus, maximum likelihood estimation was used with robust standard errors and a Satorra–Bentler corrected test statistic [28]. As the $\chi^2$ test is sensitive to sample size and complexity of the model [29], additional fit measures are reported for evaluation of the model fit. Based on recommendations [29], to define an adequate fit, the standardised root mean square residual and root mean square error of approximation should be <0.05, and the normed fit index, comparative fit index and Tucker–Lewis index should be >0.95. We performed bootstrapping [30] and applied Monte-

Carlo re-sampling techniques to test the appropriateness of the model parameters, standard errors, confidence intervals and fit indices [30, 31], because the sample size was only 57 debriefings. Finally, internal consistency was evaluated using Cronbach's alpha ($\alpha$), and composite reliability using Omega ($\omega$) [32].

## Results

### Phase 1: Validity evidence based on content

The G-OSAD was developed based on the original English OSAD, through translation, back-translation and cross-cultural adaptation, to match the content of the original version. Nine expert instructors of SBME answered the semi-structured questionnaire to provide evidence for the validity based on content. One instructor did not answer the question "How well can an instructor's ability to conduct a debriefing be assessed by the German OSAD version?". The mean score of the eight instructors was 8.9 ±1.0 on the 10-point Likert-scale. The open questions revealed that this G-OSAD is useful to assess the quality of debriefing, and has the potential to contribute to performance improvement (see S2 Appendix for examples of typical answers). Suggestions from the instructors that deviated from the basic content and scope of the original English OSAD were not considered, to not deviate significantly from the original English OSAD and endanger comparability. For example, suggestions concerning the reformulation of behavioural examples or the addition of new behavioural examples to facilitate assessment (not existing in the original English OSAD) were not adopted. This final G-OSAD then entered Phase 2.

### Phase 2: Validity evidence based on internal structure

The 14 instructors who facilitated the 57 debriefings that were used to rate the G-OSAD were on average 43.3 ±8.8 years old, with mean working experience as instructors of 4.9 ±4.2 years; three were women (21%).

The descriptive statistics of the G-OSAD ratings of the three trained raters across all of the debriefings are summarised in Table 1.

The interrater reliability (ICC) of the G-OSAD scores for each category (excluding 'establishes learning environment'; see above) are reported in Table 2. The ICC for the average ratings of the three raters was 0.848 for the total score and for the different categories ranged

**Table 1. Descriptive statistics of the G-OSAD ratings of the three trained raters across all of the debriefings (N = 57).**

| G-OSAD category | | Rater 1 | Rater 2 | Rater 3 | Mean |
|---|---|---|---|---|---|
| 1 | Approach | 3.72 ±0.67 | 3.74 ±0.52 | 3.75 ±0.51 | 3.74 ±0.51 |
| 2 | Establishes learning environment[a] | - | - | - | - |
| 3 | Engagement of learners | 4.48 ±0.82 | 4.07 ±0.82 | 4.18 ±0.85 | 4.24 ±0.73 |
| 4 | Reaction phase | 2.75 ±0.83 | 2.39 ±0.84 | 2.65 ±0.92 | 2.60 ±0.64 |
| 5 | Description phase | 4.61 ±0.82 | 3.95 ±0.77 | 4.04 ±0.76 | 4.05 ±0.65 |
| 6 | Analysis phase | 4.50 ±0.68 | 4.19 ±0.85 | 4.16 ±0.73 | 4.28 ±0.63 |
| 7 | Diagnosis phase | 4.63 ±0.67 | 4.46 ±0.60 | 4.40 ±0.73 | 4.50 ±0.56 |
| 8 | Application phase | 3.79 ±0.67 | 3.67 ±0.69 | 3.60 ±0.68 | 3.68 ±0.49 |
| **9** | **Total score** | **28.02 ±3.76** | **26.46 ±3.32** | **26.77 ±3.58** | **27.08 ±3.16** |

For understanding here, the categories of the G-OSAD are presented in English

Possible score for the different G-OSAD categories, 1 (minimum) to 5 (maximum)

[a], Category not assessed because it was not part of the debriefing videorecordings, and is therefore excluded

**Table 2. Interrater reliability (as intraclass correlation coefficient) for the single and average ratings for each category across all of the debriefings (N = 57).**

| G-OSAD category | | Intraclass correlation coefficient [mean (95% confidence interval)] | | |
|---|---|---|---|---|
| | | **Single rater** | **Two raters[b]** | **Three raters** |
| 1 | Approach | 0.678 (0.552–0.782)[2] | 0.812 (0.680–0.889)[2] | 0.863 (0.787–0.915)[2] |
| 2 | Establishes learning environment[a] | - | - | - |
| 3 | Engagement of learners | 0.624 (0.469–0.749)[2] | 0.768 (0.538–0.872)[2] | 0.833 (0.726–0.899)[2] |
| 4 | Reaction phase | 0.318 (0.158–0.485)[2] | 0.479 (0.131–0.689)[1] | 0.583 (0.360–0.739)[2] |
| 5 | Description phase | 0.534 (0.384–0.671)[2] | 0.692 (0.477–0.818)[2] | 0.775 (0.652–0.860)[2] |
| 6 | Analysis phase | 0.516 (0.358–0.659)[2] | 0.679 (0.436–0.815)[2] | 0.762 (0.626–0.853)[2] |
| 7 | Diagnosis phase | 0.540 (0.390–0.676)[2] | 0.697 (0.483–0.822)[2] | 0.779 (0.657–0.862)[2] |
| 8 | Application phase | 0.274 (0.112–0.446)[2] | 0.429 (-0.077–0.663)[1] | 0.531 (0.275–0.707)[2] |
| **9** | **Total score** | **0.650 (0.503–0.767)[2]** | **0.790 (0.604–0.883)[2]** | **0.848 (0.752–0.908)[2]** |

For understanding here, the categories of the G-OSAD are presented in English

[a], Category not assessed because it was not part of the debriefing videorecordings, and is therefore excluded

[b], Calculated as means of the three different possible pairs of raters.

[1], p <0.05;

[2], p <0.001.

between 0.531 to 0.863. The ICC for the average ratings of two raters was 0.790 for the total score and ranged between 0.429 to 0.812 for the categories. The single ratings yielded lower ICCs with 0.650 for the total score and a range of 0.274 to 0.678 for the categories (Table 2). ICC values <0.40 are considered poor, 0.40 to 0.59 fair, 0.60 to 0.74 good, and 0.75 to 1.00 excellent [33].

The test-retest reliability of the total average scores of the 10% G-OSAD ratings reassessed after 2 months showed ICC of 0.976 (95% confidence interval, 0.837–0.997; $p = 0.001$). These psychometrical characteristics are given in Table 3. The agreement between the average total G-OSAD ratings and the average total original English OSAD ratings was ICC = 0.586 (95% confidence interval, 0.297–0.756; $p = 0.001$).

For the CFA based on the average ratings of the G-OSAD categories (excluding 'establishes learning environment'; see above) the adequacy of the observed data was confirmed by the Kaiser–Meyer–Olkin criterion of 0.83, and the Bartlett's test of sphericity of $\chi^2$ (21) of 263.61 ($p < 0.001$).

**Table 3. Psychometric properties of the original English Objective Structured Assessment of Debriefing (OSAD) [17] and the German OSAD.**

| Psychometric property | Objective Structured Assessment of Debriefing | | | | |
|---|---|---|---|---|---|
| | Original | German | | | |
| | **English[a]** | **Overall** | **Single rater** | **Two raters** | **Three raters** |
| Cronbach's α | 0.89 | 0.87 (CI 0.81–0.92) | -- | -- | -- |
| Omega (ω) | n.a. | 0.93 | -- | -- | -- |
| Test-retest reliability | 0.89 | 0.98 | -- | -- | -- |
| **Interrater reliability** (ICC) | | | | | |
| Total score | 0.88 | | 0.65 | 0.79 | 0.87 |
| Range (individual categories) | n.a. | | 0.27–0.68 | 0.43–0.81 | 0.53–0.86 |

ICC, intraclass correlation coefficient; CI, confidence interval

[a], ICC of the total score only reported; single and average ratings and Omega ω not reported

n.a., data not available

Exploratory factor analysis was performed to explore the dimensional structure of the G-OSAD. The results of the parallel analysis [34], the 'very simple structure criterion' [35], the inspection of the scree plot, and Velicer's minimum average partial test [36] indicated extraction of one factor. CFA was also caried out to establish the factorial validity for the measurement model. The robust $\chi^2$ test remained significant ($\chi^2[14, N = 57] = 31.08; p = 0.005$). The fit indices for the overall fitting for the different one factor models are given in Table 4, which are acceptable according to the published guidelines [29]. Modification indices were examined to determine the areas of localised strain. No large modification indices ($>10$) were seen, and no *post-hoc* modifications were made to improve the overall model fit.

**Fitting of the internal structure.** The standardised parameter estimates range was $\beta = 0.20$–$0.94$ (see Fig 1). Thus, in the final model, two categories (i.e., 'reaction phase', $\beta = 0.200$; 'application phase', $\beta = 0.385$) were little correlated with the latent structure. The full model is as shown in Fig 1.

**Convergent validity.** The average variance extracted is an indicator of the convergent validity, and thus it defines the variance captured by the construct in relation to the variance due to measurement error [37]. The average variance extracted for the global factor was 0.52, which exceeded the suggested minimum of 0.5 [37].

**Item analysis and scale reliability.** The item discrimination *(rjt)*, internal consistency (Cronbach's α) and composite reliability (ω) of the categories of the G-OSAD were determined. The discrimination between participants of the categories with high *versus* low scores was satisfactory, with item discrimination coefficients from *rjt* = 0.60 to *rjt* = 0.88, except for the 'reaction phase' category (*rjt* = 0.44). Table 3 includes the data for internal consistency, given by Cronbach's α, and composite reliability, given by total ω.

## Discussion

This study establishes a German version of the original English OSAD [17] according to the gathered validity evidence for the assessment of the quality of the debriefing. We used the framework of validity described in the 'Standards of Educational and Psychological Measurement' [20]. The evidence based on the content and internal structure supports the G-OSAD validly for the assessment of the quality of the debriefing.

According to the framework of validity [20], the thorough multistage translation, back-translation and cross-cultural adaptation serves as a first content validity element to ensure consistency with the evidence-based original English OSAD [17]. The supporting quantitative and qualitative feedback from expert instructors of SBME regarding the adequacy of the G-OSAD content to assess the quality of the debriefings added further validity evidence based on content. In addition to the supporting validity evidence based on the content of the G-OSAD, that was gathered in the first phase, validity evidence elements based on the internal structure were collected in the second phase.

**Table 4. Fitting statistics for the one factor model for all of the debriefings (N = 57).**

| One factor model | Absolute fit indices | | | | Comparative fit indices | | |
|---|---|---|---|---|---|---|---|
| | $\chi^2(df)$ | $\chi^2/df$ | SRMR | RMSEA (90% CI) | NFI | CFI | TLI |
| Robust | 31.075 (14) | 2.22 | 0.086 | 0.146 (0.078–0.214) | 0.87 | 0.92 | 0.88 |
| Bootstrap | 15.24 (14) | 1.09 | 0.049 | 0.038 (0.00–0.179) | 0.93 | 0.98 | 0.99 |
| Monte Carlo simulation | 15.68 (14) | -- | 0.066 | 0.043 (0.039–0.047) | -- | 0.96 | 0.97 |

Df, degrees of freedom; SRMR, standardised root mean square residual; RMSEA, root mean square error of approximation; NFI, normed fit index; CFI, comparative fit index; TLI, Tucker–Lewis index

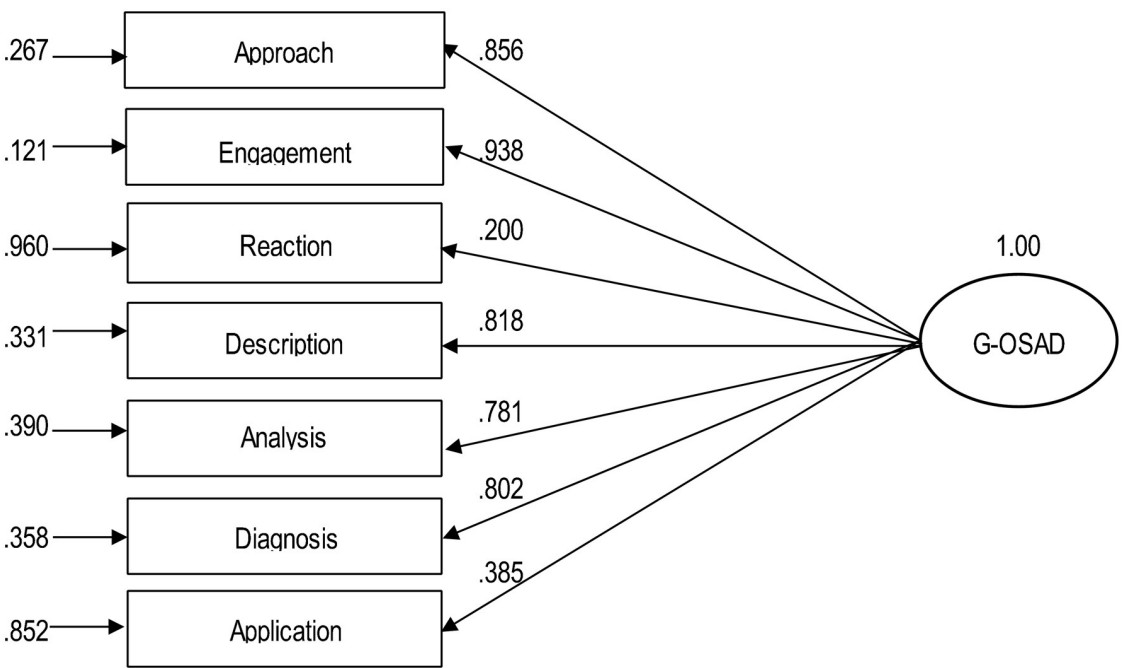

**Fig 1. Factor structure of the G-OSAD.** Full model and standardised parameter estimates. Rectangles, observed variables; oval, latent construct. The paths from the latent construct to the observed variables indicate the standardised loading (β) of each variable. The arrows to the observed variables (left) indicate the measurement errors (ε). The category of 'establishes learning environment' was not assessed because it was not part of the video recordings of the debriefings, and therefore it was excluded.

According to Cicchetti [33], the interrater reliability of the average ratings of the three raters was excellent for the total ICC score of 0.848, and for nearly all of the categories (except for 'reaction phase', 'application phase'; Table 2). The ICCs of the average ratings of two raters were good to excellent (except for 'reaction phase', 'application phase'). In contrast, the ratings of the individual raters were fair to good (again, except for 'reaction phase', 'application phase'). This indicates that the average ratings of two raters are highly reliable for the total score for the measurement of the overall debriefing quality. For the differentiated analysis of the categories of G-OSAD, average ratings of three raters are needed to obtain highly reliable data. One open question is whether more extensive rater training, content expertise or different rater populations (for example raters with clinical background or SBME instructors) might achieve highly reliable results with fewer raters.

Interestingly, the two categories of 'reaction phase' and 'application phase' yielded the lowest interrater reliabilities. Unfortunately, the report of the original OSAD [17] did not include the interrater reliability for the individual categories, only for the total score. Therefore, it is not possible to know whether the low interrater reliability of these two categories was a general issue also in the original OSAD, or whether it is due to the translation into German or the specifics of the study setting. For example, the clarity of the descriptions and behavioural anchors of the different categories could vary for raters with different backgrounds and may depend on the applied rater training. Furthermore, some categories could generally be more difficult to assess depending on the background of the raters (i.e. content expertise). Thus, further investigation of the G-OSAD incorporating different rater populations and rater training might improve the quality of the application of the G-OSAD.

Test-retest reliability of the total scores (ICC = 0.98) was even higher than the reported value for the original OSAD scores (ICC = 0.89) [17] (Table 3). We do not know whether this

difference can be attributed to the different retest periods (English OSAD, 6 month; G-OSAD, 2 months). We chose a shorter time span to reach a reasonable compromise between recollection bias and unsystematic changes in perception and behaviour of the trained student raters (i.e. uncontrolled knowledge growth due to different lectures, variations in performance due to differences in exam burden) [38]. Nevertheless, this might have led to an overestimation of the test-retest coefficients and influenced the comparability of our results to those of the English OSAD. Regrettably, the other comparable debriefing quality scoring tool, the Debriefing Assessment for Simulation in Healthcare [18] does not report any test-retest reliability. This opens a wide field for future research.

The agreement between the G-OSAD ratings and the original English OSAD ratings can be considered as fair [33]. We consider that the ratings of the debriefing quality by the G-OSAD and the original English OSAD are comparable, which further supports the successful translation.

The OSAD was developed to assess the quality of debriefing. The quality of debriefing is a construct, and CFA can be used to explore the underlying structure of the OSAD ratings. For the original English OSAD, CFA is not available, but we conducted CFA to determine the underlying structure of the ratings of the G-OSAD categories. The CFA based on the ratings of the 57 debriefings indicated a single underlying factor. As all of the categories of the G-OSAD included in the analysis could therefore be combined for this single underlying factor (Fig 1), this supports the interpretation that the construct quality of the debriefings is measured by the G-OSAD ratings.

Although the variance explained by the construct exceeded the suggested minimum at 0.52 [37], this value still indicates that a non-negligible amount of variance due to measurement error. Additionally, the scoring of the 'reaction phase' and 'application phase' were not well covered by the factor (Fig 1). As reported in Table 2, these two categories had the lowest inter-rater reliabilities. This might be the reason for the low explanation through the factor, and does not necessarily indicate that they relate less to the factor. However, this not so reliable assessment of these two categories (discussed above) can be seen as one main reason for the amount of variance due to measurement error.

The internal consistency of the G-OSAD scoring as assessed by Cronbach's α was acceptable (0.87) according to guidelines [39], and comparable to that for the original OSAD (0.89) (Table 3). This means that the raters scored the different categories of the G-OSAD similarly, which indicates that the categories are interrelated. Omega (ω) is seen as a more appropriate index of the extent to which all of the items in a scale measure the same latent variable [32]. The G-OSAD showed an excellent ω of 0.93, which supports that the ratings of the G-OSAD categories measure the same construct [32].

The three psychometric characteristics of the G-OSAD ratings of interrater reliability, test-retest reliability and internal consistency are comparable to those of the original English OSAD (Table 3). This indicates that the thorough translation process successfully maintained its consistency with the original OSAD.

The validity evidence based on content and internal structure provide support that the G-OSAD ratings assess the construct of the debriefing quality. Thus, the G-OSAD is a tool that is now available for German-speaking areas that can be used to enhance debriefing quality. As debriefings are an essential mechanism for learning in SBME [12–16], the G-OSAD might contribute to the effectiveness of SBME, which has been indicated as a concern in research in this field [2, 4].

A study limitation here might be related to the generalisability if the translation process took place in one German-speaking area. However, here, German speakers from Austria, Germany and Switzerland translated the original English OSAD into German. As with every

assessment tool, interpretations regarding the validity are associated with the investigated set-ting. For example, in our setting the debriefings were co-facilitated by two instructors, rated by trained psychology students and had a duration between 20 to 59 minutes.

Another limitation is that the category 'establishes learning environment' was not assessed and analysed. At the Bern University Hospital Simulation Centre the establishment of a con-ducive learning environment is part of the introduction at the beginning of the SBME courses [7], which is never video recorded and could therefore not be assessed. This might affect the comparability of interrater reliability, test-retest reliability and internal consistency. Further research on the G-OSAD should not only incorporate ratings of this category, but also investi-gate its validity evidence to external criteria. However, the original OSAD investigated rela-tions to the participant evaluations of satisfaction and educational value, which have already been reported [17].

## Conclusion

This study provides validity evidence based on content and internal structure for the valid assessment of the quality of debriefing by the G-OSAD. Therefore, the G-OSAD is a relevant tool to assess debriefing quality in German-speaking SBME.

## Supporting information

**S1 Appendix. German version of the OSAD (G-OSAD).**
(PDF)

**S2 Appendix. Typical questionnaire answers for the German version of the OSAD.**
(PDF)

## Acknowledgments

The authors thank Renée Gastring, Gaelle Haas and Nadja Hornburg for their help with the data collection, and Yves Balmer, Mey Boukenna, Gianluca Comazzi, Tatjana Dill, Kai Kranz, Sibylle Niggeler and Francis Ulmer for their support in the translation process. They also thank all of the instructors for their feedback on the G-OSAD. Additionally, they thank Chris Berrie for critical review of the manuscript language.

## Author Contributions

**Conceptualization:** Sandra Abegglen, Robert Greif.

**Data curation:** Sandra Abegglen, Helen Eigenmann.

**Formal analysis:** Sandra Abegglen, Helen Eigenmann.

**Funding acquisition:** Robert Greif.

**Investigation:** Sandra Abegglen, Andrea Krieg, Helen Eigenmann, Robert Greif.

**Project administration:** Sandra Abegglen, Robert Greif.

**Resources:** Robert Greif.

**Supervision:** Sandra Abegglen, Robert Greif.

**Writing – original draft:** Sandra Abegglen, Andrea Krieg, Helen Eigenmann, Robert Greif.

**Writing – review & editing:** Sandra Abegglen, Andrea Krieg, Helen Eigenmann, Robert Greif.

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
