## [Decision Letter · Decision Letter 0]

3 Dec 2020

PONE-D-20-32979

Objective Structured Assessment of Debriefing (OSAD) in simulation-based medical education: translation and validation of the German version

PLOS ONE

Dear Dr. Abegglen,

Thank you for submitting your manuscript to PLOS ONE. After careful consideration, we feel that it has merit but does not fully meet PLOS ONE’s publication criteria as it currently stands. Therefore, we invite you to submit a revised version of the manuscript that addresses the points raised during the review process.

We look forward to receiving your revised manuscript.

Kind regards,

Frantisek Sudzina

Academic Editor

PLOS ONE

Journal Requirements:

Reviewers' comments:

Reviewer's Responses to Questions

**Comments to the Author**

1. Is the manuscript technically sound, and do the data support the conclusions?

Reviewer #1: Yes

Reviewer #2: Yes

2. Has the statistical analysis been performed appropriately and rigorously? 

Reviewer #1: Yes

Reviewer #2: I Don't Know

3. Have the authors made all data underlying the findings in their manuscript fully available?

Reviewer #1: Yes

Reviewer #2: Yes

4. Is the manuscript presented in an intelligible fashion and written in standard English?

Reviewer #1: Yes

Reviewer #2: Yes

5. Review Comments to the Author

Reviewer #1: Thank you very much for a very interesting work, that will be an important contribution to the German-speaking SBME community. I have noted in the comments (directly in the document) two weaknesses of the otherwise very sound study - I am sure they can be addressed in an easy way by commenting on the questions I raised. Apart from that, I am very happy to see this valuable tool being translated into German and comprehensively analysed.

Reviewer #2: Dear authors

I read this article with interest and I think this is an important contribution in the field of simulation base training and debrief assessment.

I have some remarks and question, which could be addressed

1) You are using the wording: Simulation-based medical education (SBME), what is the difference to simulation-based medical training? And why have you used this phrase?

2) Line 107; (4) Two bilingual speakers with medical backgrounds….

What is the rationale behind 2 independent translators? Not one or several?

3) Line 127 ……debriefings (duration, 20-59 min.)….. How is the validity of the OSAD in debriefing with significant different timespans?

4) Line 201. Could you please provide more background to the 14 instructors

5) Table 1: Could you please provide the possible maximum score for the different G OSAD categories

6) Table 2 The presentation of the different p values should be mor clear. Maybe you can find another table presentation?

7) Line 270 Could you please discuss and present the results of the convergent validity in the limitations?

8) Line 315; why have you chosen a short timespan for the examination of the test-retest reliability?

6. PLOS authors have the option to publish the peer review history of their article (what does this mean?). If published, this will include your full peer review and any attached files.

Reviewer #1: **Yes: **Dr. Marc Lazarovici

Reviewer #2: No

---

## [Author Response · Author response to Decision Letter 0]

16 Dec 2020

We wish to thank the editor and the reviewers for the valuable comments and suggestions, and for this opportunity to revise our manuscript. We adressed all questions in the Response to Reviewers letter. We hope with these changes and adaptation our manuscript is now suitable for publication in PLOS ONE and look forward to receiving your reply.

---

## [Editor Report · Decision Letter 1]

17 Dec 2020

Objective Structured Assessment of Debriefing (OSAD) in simulation-based medical education: translation and validation of the German version

PONE-D-20-32979R1

Dear Dr. Abegglen,

We’re pleased to inform you that your manuscript has been judged scientifically suitable for publication and will be formally accepted for publication once it meets all outstanding technical requirements.

Kind regards,

Frantisek Sudzina

Academic Editor

PLOS ONE
---

## [Editor Report · Acceptance letter]

21 Dec 2020

PONE-D-20-32979R1 

Objective Structured Assessment of Debriefing (OSAD) in simulation-based medical education: translation and validation of the German version 

Dear Dr. Abegglen:

I'm pleased to inform you that your manuscript has been deemed suitable for publication in PLOS ONE. Congratulations! Your manuscript is now with our production department. 

Kind regards, 

on behalf of

Dr. Frantisek Sudzina 

Academic Editor

PLOS ONE